# Optogenetic inhibition-mediated activity-dependent modification of CA1 pyramidal-interneuron connections during behavior

Igor Gridchyn[†], Philipp Schoenenberger[†‡], Joseph O'Neill[§], Jozsef Csicsvari*

Institute of Science and Technology Austria (IST Austria), Klosterneuburg, Austria

**Abstract** In vitro work revealed that excitatory synaptic inputs to hippocampal inhibitory interneurons could undergo Hebbian, associative, or non-associative plasticity. Both behavioral and learning-dependent reorganization of these connections has also been demonstrated by measuring spike transmission probabilities in pyramidal cell-interneuron spike cross-correlations that indicate monosynaptic connections. Here we investigated the activity-dependent modification of these connections during exploratory behavior in rats by optogenetically inhibiting pyramidal cell and interneuron subpopulations. Light application and associated firing alteration of pyramidal and interneuron populations led to lasting changes in pyramidal-interneuron connection weights as indicated by spike transmission changes. Spike transmission alterations were predicted by the light-mediated changes in the number of pre- and postsynaptic spike pairing events and by firing rate changes of interneurons but not pyramidal cells. This work demonstrates the presence of activity-dependent associative and non-associative reorganization of pyramidal-interneuron connections triggered by the optogenetic modification of the firing rate and spike synchrony of cells.

**\*For correspondence:**
jozsef.csicsvari@ist.ac.at

[†]These authors contributed equally to this work

**Present address:** [‡]School of Psychology, Cardiff University, Cardiff, United Kingdom; [§]F. Hoffmann-La Roche Ltd, Roche Innovation Center Basel, Grenzacherstrasse, Switzerland

**Competing interests:** The authors declare that no competing interests exist.

## Introduction

It is increasingly recognized that plastic reorganization of brain circuits needed for learning and other cognitive functions involves not only principal cells but also local inhibitory interneurons (*Buzsáki, 2010*). A large body of work has demonstrated that excitatory connections onto inhibitory interneurons, as well as inhibitory connections on principal cells, are often plastic (*Kullmann and Lamsa, 2007*; *McBain and Kauer, 2009*). However, the rules governing the plasticity of excitatory synapses on inhibitory interneurons are not always similar to that targeting other principal cells (*Bartos et al., 2011*; *Lamsa et al., 2007*; *Pelkey et al., 2017*). Even in the CA1 region of the hippocampus, plasticity rules can be different, depending on the experimental conditions and fibers stimulated. Although in the majority of cases some pyramidal-interneuron cell connections show anti-Hebbian non-associative plasticity, others show weight changes that are governed by Hebbian rules (*Le Roux et al., 2013*; *Nissen et al., 2010*; *Topolnik et al., 2009*). In addition, some inhibitory interneuron types in the hippocampus do not seem to possess plastic synapses with their excitatory inputs, such as the CCK cells (*Nissen et al., 2010*).

In vivo work, primarily during anesthesia, has also demonstrated that plastic alterations can occur between afferent excitatory fibers and CA1 interneurons (*Buzsáki and Eidelberg, 1982*; *Lau et al., 2017*). This work showed that stimulation of these fibers could either up- or downregulate evoked spike responses, depending on the interneuron subtype. However, our knowledge about the precise reorganization of pyramidal-interneuron connections during behavior is limited because of the technical challenge of directly performing patch-clamp recordings from monosynaptically connected pyramidal cell-interneuron pairs during such conditions. It is, however, possible to study these

connections indirectly by identifying monosynaptically connected pyramidal cell-interneuron pairs by using cross-correlation analysis of the spike timing and measuring spike transmission probability between them (*Csicsvari et al., 1998*; *Csicsvari et al., 2003*; *Marshall et al., 2002*). Early work demonstrated the behavioral state-dependent modulation of such connections (*Csicsvari et al., 1998*). Moreover, changes in spike transmission probability have been seen in the prefrontal cortex during behavioral tasks (*Fujisawa et al., 2008*). In the hippocampus, spatial learning can cause lasting changes in these connections (*Dupret et al., 2013*). While such studies provide strong evidence of plasticity at excitatory-interneuron synapses, so far, no data has established a causal link between pre- and postsynaptic firing to changes in connection strength during behavior.

In this study, we optogenetically interfered with the circuit function by activating Halorhodopsin or Archaerhodopsin in a subpopulation of pyramidal cells and interneurons. This manipulation led to the inhibition of a subset of pyramidal cells and interneurons and also light-triggered disinhibition of many pyramidal cells (*Gridchyn et al., 2020*; *Schoenenberger et al., 2016*). Here, we examined whether these light-induced rate changes and the associated network effects could lead to the lasting reorganization of pyramidal-interneuron connection weights, as assessed by monosynaptic spike transmission probabilities.

## Results

We recorded multiple unit and field potential activities from the dorsal hippocampus in five rats, during exploration and quiet immobility sessions. In these rats, Halorhodopsin (NpHR-YFP, n = 4 rats) or Archaerhodopsin (ArchT, n = 1 rat) was expressed in the dorsal CA1 region of the hippocampus under the control of the CaMKIIα promoter using an adeno-associated virus (AAV2/1). In four rats, fifteen independently-movable tetrodes and one 200 μm optic fiber centered in the middle of the tetrode bundle targeted the dorsal CA1 region while, in the remaining animal, 24 tetrodes and four optic fibers were used (see Materials and methods). We recorded during four 25 min exploration sessions in which first a familiar environment (FAM1) was explored followed by a novel environment (NOV), and finally, the animals were returned to the familiar environment for the remaining two sessions. During the second familiar exploration session laser stimulation was applied (FAML) in a fixed part of the environment but not in the last exploration (FAM2, see *Figure 1A*). We tested whether the light application affected the behavior of the animals (*Figure 1—figure supplement 1*). In all sessions, neither the average speed nor the occupancy within the light stimulation sector were significantly different, compared to the part of the environment where no light was triggered (all p>0.5607). We identified monosynaptically-connected pyramidal cell-interneuron pairs by calculating the cross-correlation of their spike firing times and testing for the presence of a short-latency (1–2 ms) sharp (1–2 ms wide) peak. This peak indicates the presence of a monosynaptic connection in which the presynaptic pyramidal cell can discharge the postsynaptic interneuron within a short latency. In turn, the magnitude of the peak, that is, its transmission probability, reflects the connection weight between a given cell pair. Changes in firing rate across sessions by either or both cells in the pair would result in an alteration of the magnitude of the peak that does not reflect a change in this connection weight, but rather a general change in the probability of joint firing. To account for this, throughout all the analysis, we measured the chance probability that the pair fires together, by averaging the correlation probabilities over the 10–50 ms time bins and subtracting this from the peak. Altogether, we identified 78 pyramidal cell-interneuron pairs in these recordings (see Materials and methods).

The light stimulation inhibited not only a selected population of pyramidal cells but also many interneurons, while a further group of pyramidal cells increased their firing due to disinhibition, as assessed directly by their light responses to brief light pulses in the rest session at the end of the recordings (*Figure 1B*). In our previous work, we showed that these disinhibited pyramidal cells only increased their firing after the maximum light-mediated suppression on interneurons (*Gridchyn et al., 2020*; *Schoenenberger et al., 2016*). We also showed before (*Schoenenberger et al., 2016*) that both somatostatin- and parvalbumin immunopositive interneurons can express transgenes following AAV-mediated transduction in agreement with earlier work (*Nathanson et al., 2009*). It is possible, however, that other adeno-associated virus serotypes or the usage of the same virus in other brain regions may yield principal cell-specific expression. As a result, many pyramidal cell - interneuron pairs with monosynaptic connections showed changes in firing

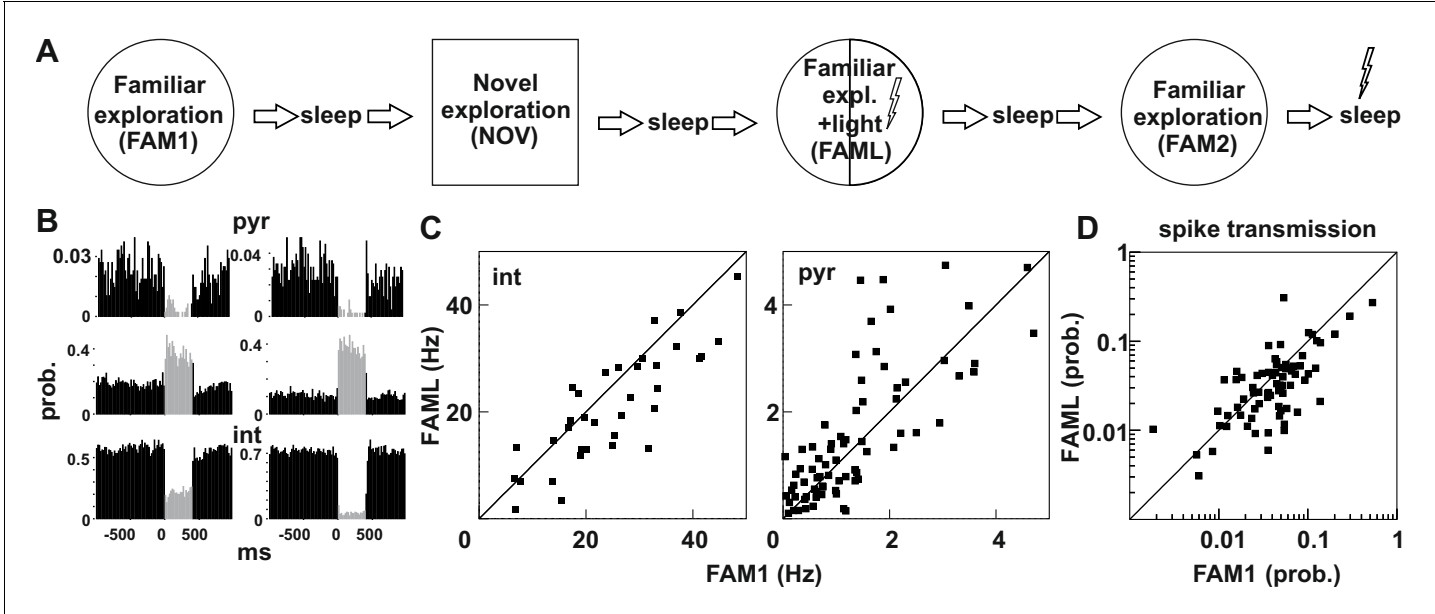

**Figure 1.** Light-induced firing changes in the CA1 region by halorhodopsin-mediated inhibition of a subgroup of pyramidal cells and interneurons. (**A**) Experimental paradigm: on each recording day the animal was exposed three times to the same familiar environment (FAM1, FAML, and FAM2), including one session in which light stimulation was triggered (FAML) as the animal explored a defined sector of the environment ($^1/_3$ – ½ of the arena). In addition, the animal also explored a novel environment (NOV). Each behavioral session was flanked by sleep, with 500 ms light pulses given in the last. (**B**) examples of cells in which light application suppressed activity and triggered an elevated rate, through disinhibition. int, interneuron; pyr, pyramidal cells. Light responses were measured during the last rest session by applying 500 ms test light pulses. The histograms show the probability of spiking within the 20 ms time bins. (**C**) The mean firing rate of the postsynaptic interneurons (left) and presynaptic pyramidal cells (right) that were part of a detected monosynaptic cell pair were plotted during FAM1 vs. FAML sessions. Lines represent the x = y line. Note that the majority of interneurons were inhibited by the light, whereas several pyramidal cells exhibited either prominent suppression or excitation of their rate. (**D**) Monosynaptic spike transmission probabilities also exhibit alterations during the FAML session with more cell pairs showing a reduction of spike transmission probabilities relative to FAM1.

The online version of this article includes the following figure supplement(s) for figure 1:

**Figure supplement 1.** Effect of light application on the behavior of animals.

rate during the FAML session, when compared to FAM1 (*Figure 1C*). The altered network activity during the light session is demonstrated by significantly lower correlation of the FAM1 vs FAML rates as compared to rates measured in alternating 5 s time windows within FAM1, both for pyramidal cells and interneurons (all p<0.0001, Z-test). However, no significant differences were found in the median of FAM1 and FAML firing rates (interneuron p=0.3213 pyramidal cell p=0.1448, Mann-Whitney test). The spike transmission probability of these cell pairs was also changed (*Figure 1D*). As for firing rates, the correlation FAM1 vs FAML spike transmission values was lower than that measured in alternating 5 s time windows within FAM1 (p<0.0001, Z-test). Moreover, there was a significant reduction in the median of the spike transmission probabilities from FAM1 to FAML (p<0.01. Mann-Whitney test). Because in the spike transmission measurements the chance probability that cells randomly fire together was compensated for, light-induced network modifications altered the ability of the pyramidal cell to drive the postsynaptic interneuron, beyond that of the firing rate alterations-mediated changes. Changes in connection weight between cell pairs during the FAML session could be either transient or reflect longer-term plasticity that outlasts optogenetic stimulation. Moreover, connection strength could change as place cells remap their place fields during exploration of a different environment (*Wilson and McNaughton, 1993*). We, therefore, tested whether significant changes in the spike transmission of monosynaptic pairs can be seen across sessions, relative to the baseline identified in FAM1 (*Figure 2A–B*). To do this, we generated a score that represented the absolute value of normalized spike transmission differences between sessions (difference divided by the sum, see Materials and methods). Overall, this score was the largest between NOV-FAM1 and FAML-FAM1 sessions. However, while the changes between FAM2-FAM1 sessions were about 30%

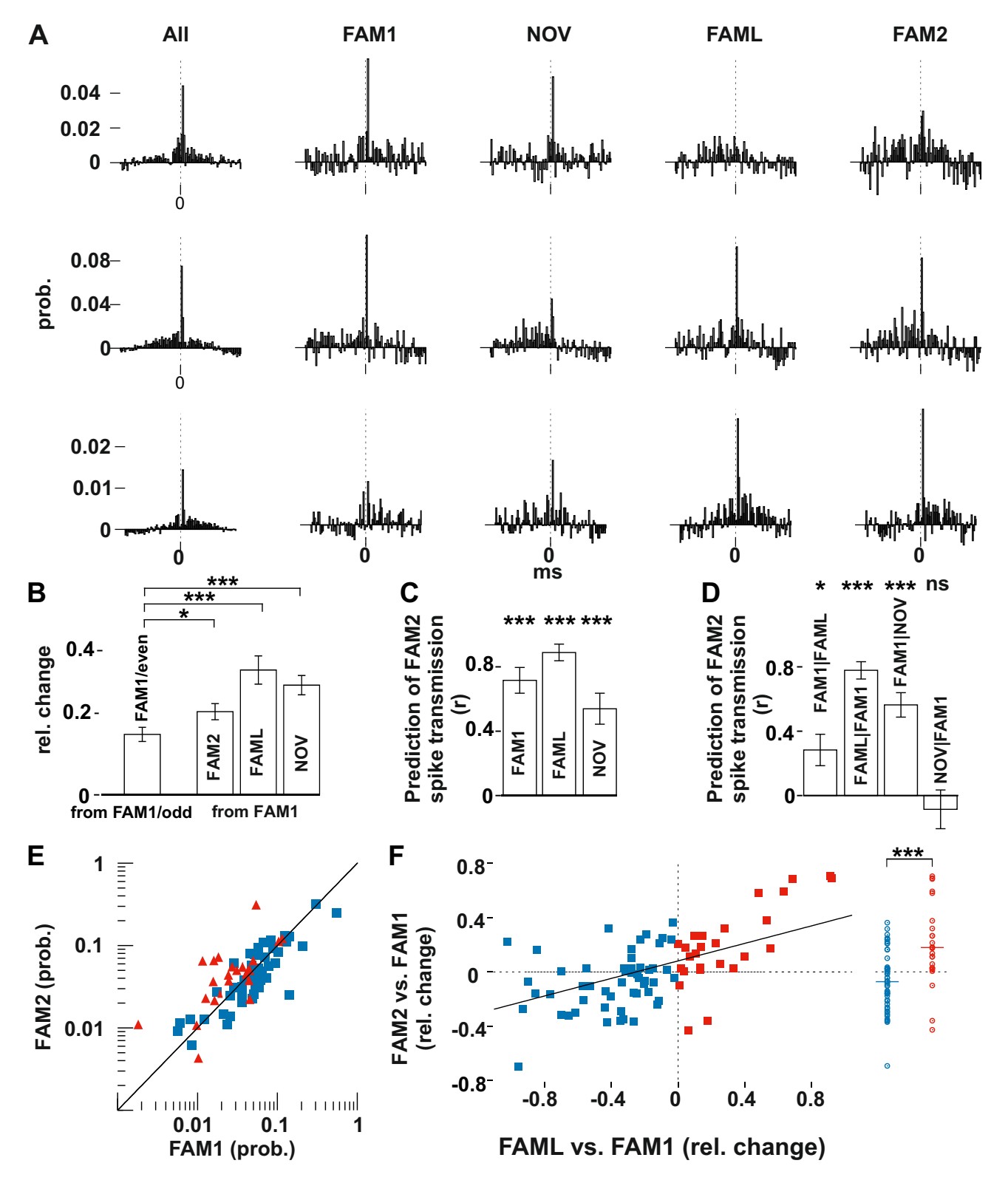

**Figure 2.** Light application triggered lasting changes in spike transmission probabilities in the same environmental context. (A) Representative examples of monosynaptic cross-correlations demonstrating altered spike transmission probabilities across different sessions. Left histograms show the cross-correlations calculated during the entire recording session (all), which were used to detect the monosynaptic pairs. The remaining histograms show the cross-correlations at different sessions. Chance joint firing probability was estimated by the average cross-correlation values in the ±10–50 ms

*Figure 2 continued on next page*

*Figure 2 continued*

bins and subtracted. Each bin represents a 1 ms time windows in [−50 ms, +50 ms] intervals. (B) Mean (± SEM) absolute difference of spike transmission probabilities, measured as relative change (difference/sum) between the odd- and even-numbered 5 s intervals of the FAM1 session and between FAM1 and other sessions. Mean (± SEM) absolute difference of spike transmission probabilities, measured as relative change (difference/sum), relative to the first FAM1 session. Note the significant reorganization of the spike transmission probabilities across all sessions, with FAM2-FAM1 being the weakest. *p=0.0268, ***p<0.0001. (C) Prediction (i.e., correlation) of transmission probabilities in FAM2 with those in the previous exploration sessions ***p<0.0001. (D) Partial correlations to illustrate the influence of each session on FAM2, while removing the effect of other behavioral sessions. Note that the linear mixed model comparison analysis showed that the NOV session did not influence FAM2 spike transmission when the FAM1 spike transmissions were taken into consideration, whereas FAM1 did influence FAM2. Significance for linear mixed model comparison is indicated. *p=0.0105, ***p<0.0001, ns not significant. (E) Spike transmission values plotted in the FAM1 and FAM2 sessions. Cell pairs that increased (red) and decreased (blue) their spike transmission in the FAML relative to FAM1 are displayed separately. Diagonal line: x = y. (F) Relative (difference/sum) changes of spike transmission probabilities between FAML-FAM1 predict those of FAM2-FAM1 changes. The relative FAM2-FAM1 changes of cell pairs with reduction (blue) and increase (red) in spike transmission from FAM1 to FAML are also displayed along a single line on the right to illustrate the negative and positive bias of these groups. The solid diagonal line represents the regression line for the data. Horizontal line: median.

weaker relative to the others, they were still significantly larger than zero (all p<0.001; ANOVA). In addition, changes between FAM2-FAM1 sessions were significantly larger than changes within FAM1 sessions as assessed by correlations measured in alternating 5 s time windows (p<0.0268, F-test), independent of the variability across animals (p=0. 0562, Likelihood-ratio test).

These population changes suggest that pyramidal cell-interneuron synaptic weights were, to some degree, reorganized by both exposure to a new environment and artificially by light stimulation. However, connection strengths were different between the first and last exposure to the same familiar environment (FAM1 and FAM2), indicating that a more lasting change had also occurred. We set out to determine the factors that accounted for the connection strength observed in FAM2 by testing at the population level whether spike transmission in FAM2 was predicted by that observed in the previous sessions. In addition, we controlled for the possible variability across animals by including animal identity as a variable into the analysis (*Figure 2C–D*). Spike transmissions in FAM2 were strongly predicted by the observed connection weights in all the previous sessions (all p<0.0001, F-test, *Figure 2C*), indicating that connection strength between the pairs was only partially reorganized. The variability across animals did not significantly account for variability in the spike transmission (all p>0.5656, Likelihood-ratio test). We then used a linear model comparison to reveal which behavioral sessions best explained the monosynaptic connection strengths in FAM2 (*Figure 2D*). We found that both FAM1 and FAML predicted FAM2 spike transmission, independent of the other (all p<0.0105, F-test). However, the model comparison showed that the NOV no longer predicted FAM2 when the effect of FAM1 was taken into consideration (p=0.4453, F-test) nor the variation across animals contributed (all p>0.5656, Likelihood-ratio test). Thus, the changes in connection weights during NOV did not influence the subsequent weights observed in FAM2, which instead was explained by weights in both FAM1 and FAML, independently. Therefore, while spike transmission values were more similar within the same environmental context, light application significantly biased spike transmission and caused lasting changes in FAM2 relative to FAM1.

The significant influence of the FAML session on FAM2 suggests that the light-induced alterations of the network activity led to lasting changes in the spike transmission probabilities even in the absence of light in the same environmental context. Direct optogenetic inhibition of some cells and the indirect firing increase of others led to either an increase or a decrease of spike transmission probabilities during the light application. Therefore, next, we tested whether the direction of change in spike transmission in FAML relative to FAM1 predicted similar change from FAM1 to FAM2 (*Figure 2E–F*). Indeed, those cell pairs in which light application led to a decrease in spike transmission relative to FAM1 also maintained weaker spike transmission values in FAM2, while cell pairs with light-enhanced transmission showed a persistent increase in transmission probability in FAM2 (all p<0.0001, F-test). Moreover, the relative change of spike transmission (difference divided by the sum) between FAML-FAM1 predicted the score change from FAM1 to FAM2 (p<0.0001, F-test), indicating that larger relative changes in spike transmission between FAML-FAM1 were accompanied by similarly larger changes between FAM2-FAM1. Thus, light-induced changes in neuronal firing during FAML may lead to lasting modifications of spike transmission. The variability across

animals did not significantly account for variability in the transmission probability change (p>0.3046, Likelihood-ratio test).

The light application can change the firing rate of the presynaptic pyramidal cell or the postsynaptic interneuron, which may, in turn, account for the observed plastic changes in transmission probability. Therefore, to address whether our effects reflected a pre- or postsynaptic mechanism, we assessed the relationship between firing rate changes within connected pairs and the modification of their monosynaptic connection. To do this, we calculated the relative firing rate changes between FAML-FAM1 for both pyramidal cells and interneurons, a measure that reflects the influence of light on the baseline firing of these cells. We then analyzed whether this measure predicted changes in transmission probabilities between FAM1 and FAM2, which reflects the longer-lasting change in synaptic strength. As a control, we also measured relative firing rate changes between FAM2-FAM1 because rate alterations reflecting the alterations in the average excitatory inputs cells received. Such changes in excitability may have influenced spike transmission beyond the rate alteration-mediated changes of the chance joint firing probability, which later were already compensated for by normalizing the histograms (*Figure 3A–B*). We found that interneuron rate changes of both FAML-FAM1 and FAM2-FAM1 influenced FAM2-FAM1 spike transmission changes, even when each other's influence was taken into account (all p<0.0064, Likelihood-ratio test). However, pyramidal cell rate changes during FAML no longer significantly influenced FAM2-FAM1 spike transmission changes when FAM2-FAM1 rate change was taken into account (p=0.0999, Likelihood-ratio test). This suggests that changes in the excitability of interneurons during the light application, as assessed by rate changes, influenced the spike transmission strength subsequently in FAM2 even when the FAM2-FAM1 rate alterations of the interneuron were compensated for. In these models, the variability across animals did not significantly account for variability in the transmission probability change (all p>0.0907, Likelihood-ratio test). In addition to the changing firing rate, the light application can cause remapping in a subpopulation of cells (*Schoenenberger et al., 2016*). However, a change in spike transmission in FAM2 did not predict the degree of pyramidal place field remapping (p=0.1612, F-test).

The finding that interneuron rate change between FAM2-FAM1 itself independently influenced FAM2-FAM1 spike transmission changes suggests that the excitability of the postsynaptic interneuron has a strong influence on the strength of spike transmission. Spike transmission changes we detected in the NOV session may, in part, have been caused by the excitability change of the interneurons. To test whether rate changes of pyramidal cells or interneurons in NOV influenced spike transmission in FAM2, we calculated normalized (difference divided by the sum) spike transmission and rate changes, in order to predict weight change separately for both cell types (*Figure 4*). As before, only interneuron rate changes predicted spike transmission changes, even when the pyramidal rate change was taken into consideration (interneuron: p<0.0001; pyramidal: p=0.1672, Likelihood-ratio test). A similar analysis was performed for the FAML session itself, where again, we found that only interneuron rate alterations predicted spike transmission alterations during the light application, even when each-others' contribution was considered (interneuron: p<0.0001; pyramidal: p=0.5885, Likelihood-ratio test). In these models, the variability across animals did not significantly account for variability in the transmission probability changes in NOV-FAM1 and FAML-FAM1 (all p>0.382, Likelihood-ratio test).

Next, we examined whether the direction of firing rate change during light application influenced the spike transmission change between FAM1 and FAM2 (*Figure 3C–D*). As before, we considered FAM1 as a baseline and generated a normalized score for the rate and weight change. Interneurons that exhibited elevated or reduced firing rate during FAML session exhibited significantly different changes in monosynaptic spike transmission probability across FAM2 and FAM1 (p<0.0001, F-test) independent of the variability across animals (p=0.3372, Likelihood-ratio test). Pyramidal cells did not show such a relationship (p=0.4499, F-test) and the variability across animals did not influence this result (p=0.1359, Likelihood-ratio test). Therefore, suppressed interneurons tended to weaken their monosynaptic weights with presynaptic pyramidal cells, whereas those that increased their rate exhibited increased weights. These effects lasted after the light application in the same environmental context. To confirm that the observed changes in monosynaptic weight were indeed mediated by a postsynaptic change in the interneuron firing rate, we differentiated monosynaptic cell pairs into four groups, according to whether the pair exhibited a rate increase or decrease, of both pyramidal cells and interneurons (*Figure 5*). A two-way ANOVA analysis showed that the interneuron

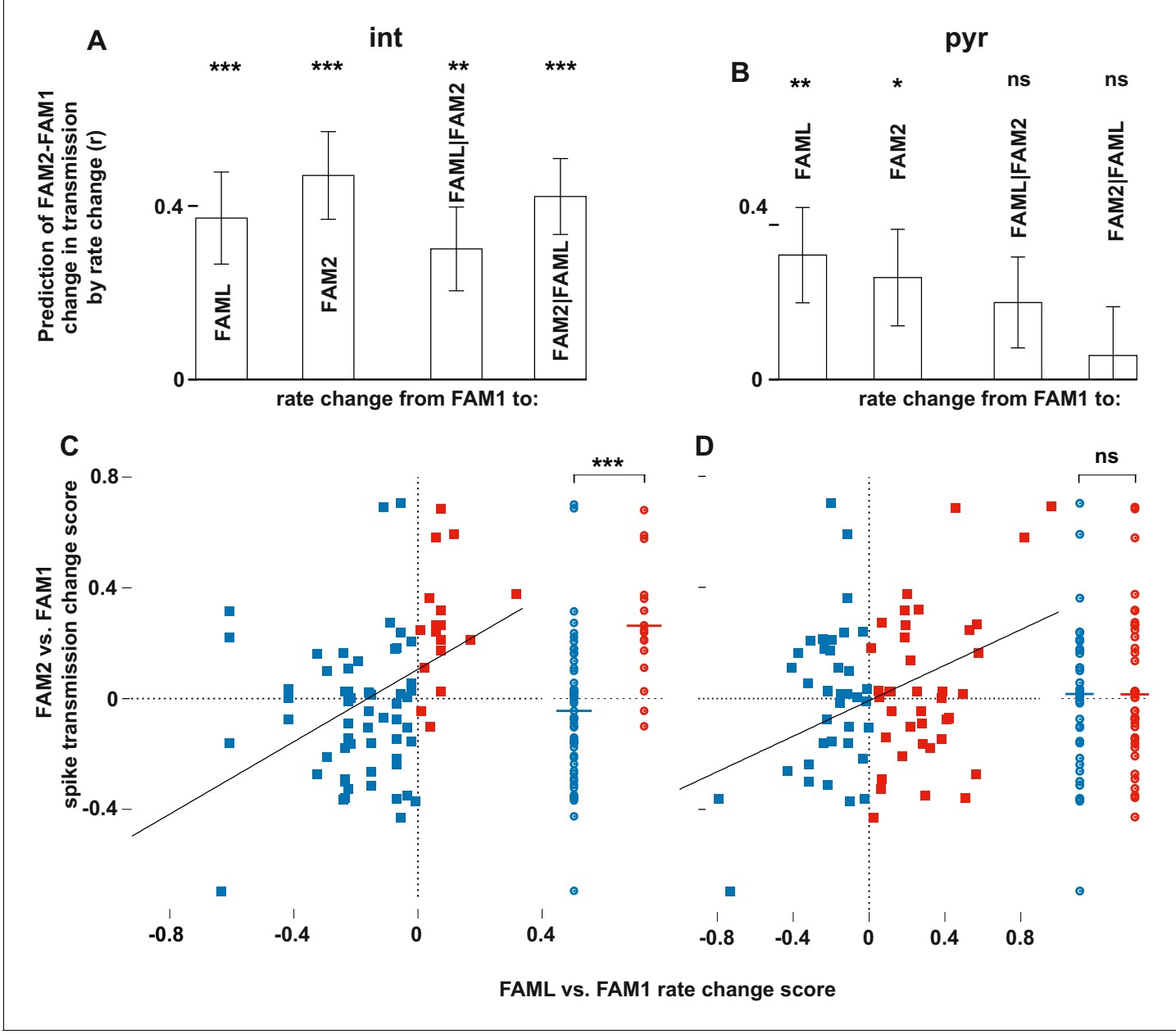

**Figure 3.** Light-induced firing rate changes of interneurons but not pyramidal cells influenced lasting familiar environment-associated spike transmission alterations between before and after the light application session. The relative changes in rate and transmission probabilities are expressed as a score throughout (difference/sum). The influence of light-induced firing rate changes on spike transmission alterations between before and after the light application session. The relative changes in rate and transmission

probabilities are expressed as a score throughout (difference/sum). (A) The correlation predicts relative FAM2-FAM1 spike transmission changes based on relative rate changes of FAML-FAM1 and FAM2-FAM1 sessions. Both correlations (left) and partial correlations (right) are shown. The comparisons of linear mixed models with one or both rate change variables show that interneuron rates in both FAML and FAM2 independently influence FAM2-FAM1 spike transmission changes. (B) same as (A) but for pyramidal cells. In this case, FAML but not FAM2 rates independently predict FAM2-FAM1 spike transmission changes. (C) Relative FAML-FAM1 rate change of interneurons versus the relative spike transmission probability changes (FAM2-FAM1) with their presynaptic pyramidal partner. Right plots spike transmission changes were plotted for the rate decrease (blue) and increase (red) pairs. Horizontal lines: median. Note that almost all pairs that exhibited an interneuron rate increase during FAML also increased their spike transmission in FAM2 and the rate decrease group exhibited a significantly smaller spike transmission change than the rate increase group. (D) Same as (C) but for pyramidal rate changes. Here the direction of rate change does not predict spike transmission changes. *p<0.0256, **p<0.0068, ***p<0.0001, ns not significant.

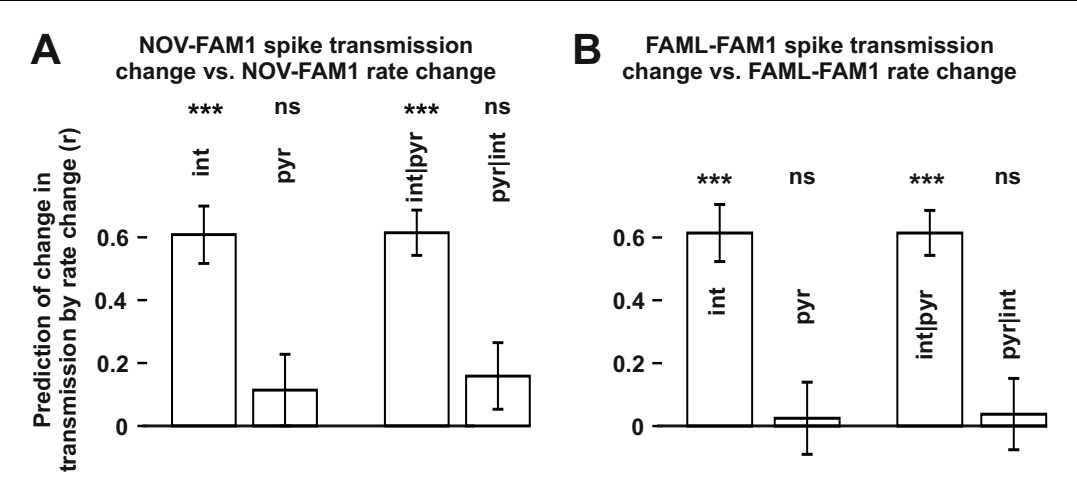

**Figure 4.** The influence of interneuron rate change on spike transmission changes in the NOV and FAML sessions, relative to FAM1. (**A**) Left: correlation of pyramidal and interneuron relative rate changes (NOV-FAM1) and the corresponding relative spike transmission changes. Partial correlations are also shown on the right. In both cases, interneuron rate changes predict the corresponding spike transmission changes but not pyramidal cells according to linear mixed model comparison. (**B**) same as (**A**) but comparing FAML-FAM1. Interneuron rate changes had a strong influence on spike transmission changes. All relative rates and transmission changes are measured as difference/sum. ***p<0.0001, ns not significant.

increase and decrease groups were significantly different, independent of the pyramidal increase and decrease as a factor (p<0.0004, F-test), but not the pyramidal group (p=0.4067, F-test), and the variability across animals did not influence the result (p=0.2749, Likelihood-ratio test) and no significant interactions were seen between cell pair groups (p=0.6109, F-test).

We found that the modulation of interneuron activity during light stimulation or exposure to a novel environment directly influenced changes in transmission probabilities between putative monosynaptic connections of pyramidal-interneuron cell pairs. This raises the possibility that such changes

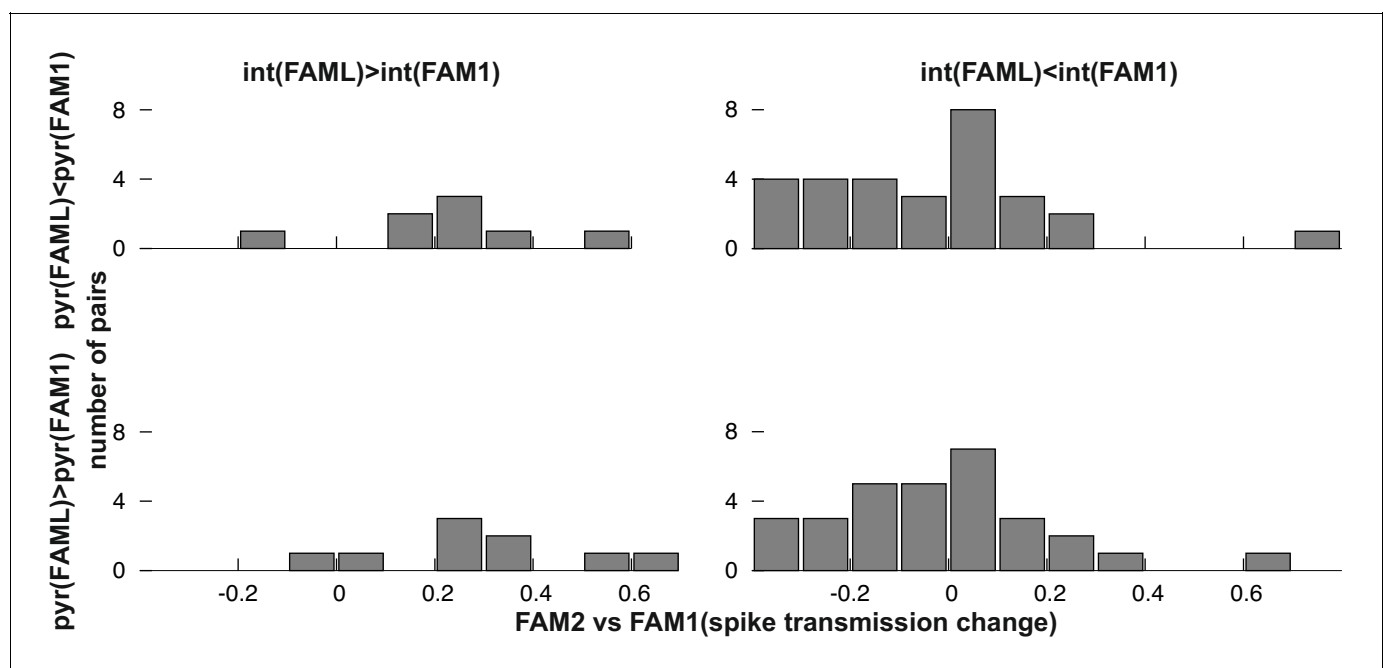

**Figure 5.** Frequency distribution of relative FAM2-FAM1 spike transmission changes for monosynaptic cell pairs according to the direction of change of the pre- and postsynaptic cell partner.

are activity-dependent. To examine this, we measured the number of instances in which pyramidal cell firing occurred in 10 ms, 20 ms, 50 ms, and 100 ms time windows before or after the interneuron spike as well as their sum for all spike-pairing events (*Figure 6*). The relative change (difference/sum) of spike pairing events was calculated between FAML and FAM1 and these spike pairing changes predicted significantly FAM2-FAM1 spike transmission change in all three cases for 50 ms window ($p < 0.00099$, F-test) even when multiple testing correction was performed while the variability across animals did not influence the result (all $p > 0.3913$, Likelihood-ratio test). In the other tested time intervals, the correlations were not significant (all $p > 0.0517$, F-test). Moreover, spike pairing event number no longer predicted the FAM2-FAM1 spike transmission changes when interneuron and pyramidal changes were together taken into account, (all $p > 0.417$, F-test). However, spike pairing itself was strongly predicted (all $R^2 > 0.859$) by pyramidal and interneuron firing rates, explaining why spike transmission changes could not be predicted independently from the combined interneuron and pyramidal rates by spike pairing.

## Discussion

Here we showed that light-induced, optogenetic alterations of the CA1 network activity can trigger lasting alterations of the monosynaptic spike transmission probability of pyramidal-cell interneuron pairs. During the light session, both changes in the postsynaptic interneuron rates and the pairing probability of cell firing predicted the changes in monosynaptic spike transmission within the same familiar environment, when sessions before and after the light interference were compared. In addition, we observed spike transmission changes in the novel environment relative to the familiar environment, which took place before the light application. These changes were specific to the novel environment, however, and were not maintained during the subsequent familiar sessions. This

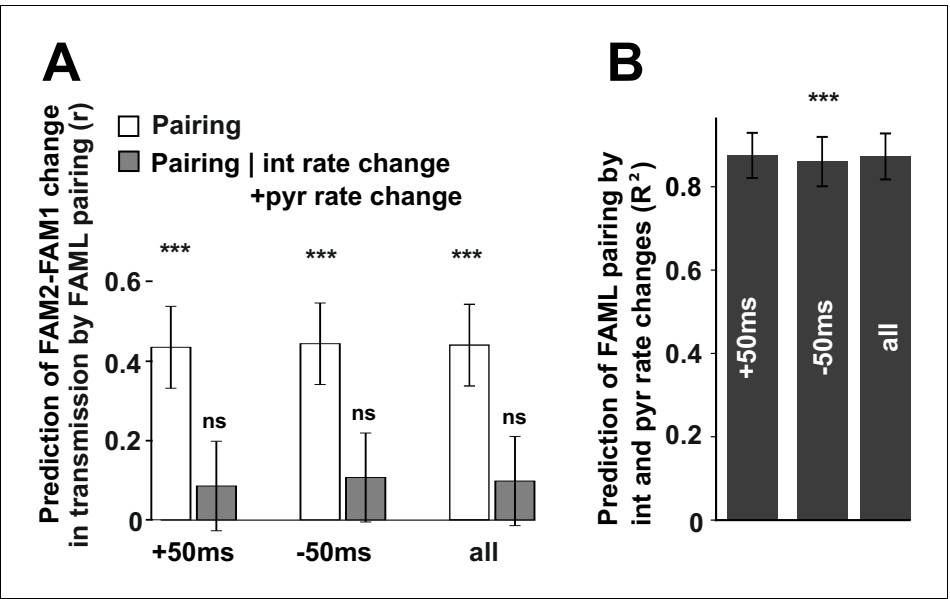

**Figure 6.** The influence of pyramidal cell-interneuron pairing on spike transmission changes. (A) The number of pairing events in FAML predicted FAM2-FAM1 spike transmission changes. The number of spike pairing events were measured in cases when interneuron spike followed by pyramidal spike within 50 ms (+50 ms) and those where it preceded that (−50 ms) and the sum of both events (all). The relative difference (difference divided by the sum) of pairing event numbers between FAML and FAM1 was calculated. Pairing change predicted with relative spike transmission change (difference divided by the sum, unfilled white histograms on the left panels). We also examined whether the light-induced interneuron and pyramidal firing rate changes that itself altered the number of pairing events alone can explain this prediction. The number of pairing events no longer predicted spike transmission changes when both pyramidal and interneuron rate changes were taken into account according to the linear mixed model comparison. (B) Change in the number of spike pairing events strongly predicted the change in interneuron and pyramidal firing rates. $R^2$ values and their 95% confidence intervals are plotted. ***$p < 0.00099$, ns not significant.

suggests that altered interneuron firing rate and the activity-dependent alterations of the pyramidal-interneuron spike pairing can modify pyramidal-interneuron connection weights during exploratory behavior. Our study did not use control animals in which only YFP was expressed. Therefore, we cannot exclude the possibility that optogenetic channel expression, or, perhaps, light application enhanced the plasticity on pyramidal-interneuron synapses. Yet, we observed similar activity-dependent changes during spatial learning before (*Dupret et al., 2013*). So, it is likely that the optogenetic, light-mediated rate alteration was a primary driver of the activity-dependent, lasting connection weight changes.

We used a measure of spike transmission probability that compensated for the changes in the chance probability that the two cells fire together as a result of firing rate alterations. However, the average depolarization level of the cell, as reflected by its mean firing rate, can lead to more efficient spike transmission, even without changes in the synaptic weight. Indeed, the spike transmission changes from FAM1 to FAM2 were influenced by the rate changes of the postsynaptic interneuron between these sessions. In turn, this suggests that the postsynaptic interneuron's general level of depolarization can influence spike transmission. However, in a similar manner, changes in FAML-FAM1 interneuron rate also predicted FAM2-FAM1 spike transmission changes. Indeed, when FAML interneuron rate increased, a stronger spike transmission was seen subsequently in FAM2, while reduced spike transmission was associated with reduced interneuron rate. Considering that the interneuron rate increase in FAML was not directly mediated by the light, we cannot exclude the possibility that stronger pyramidal inputs caused the interneuron rate increase in FAML that is caused by the plastic strengthening of these connections. Nevertheless, the excitability of interneurons in FAM2 alone, which may be indicative of plasticity-mediated input to interneurons, did not explain the spike transmission changes. Indeed, rate changes in FAML could predict spike transmission changes in FAM2-FAM1 independent of FAM2-FAM1 rate changes. That is, FAML-FAM1 rate changes predicted FAM2-FAM1 spike transmission changes even when the FAM2-FAM1 interneuron rate changes were accounted for. Therefore, FAML-FAM1 interneuron rate changes had further predictive value beyond those observed by FAM2-FAM1 rates changes and consequently excitability/depolarization alterations in FAML that were no longer present in FAM2 still predicted FAM2-FAM1 spike transmission changes. This finding indicates that interneuronal excitability changes during light application session caused lasting changes of the pyramidal interneuron connections, beyond any lasting nonspecific excitability changes that occurred between FAM2-FAM1.

We observed changes in spike transmission between FAM1 and the novel environment, which were larger in amplitude than those across the familiar environment before and after the light application. Exposure to the novel environment nevertheless did not influence changes in the familiar environment. Can we expect that pyramidal-interneuron weights change from one environment to another but they revert to the previous configuration when the animal is returned to the first environment? Although we cannot exclude this possibility, it is more likely that the average depolarization level of each interneuron is different across different environments, which in turn reveals different monosynaptic connections and connection strengths. Interneuron rates reorganize across the familiar and novel environment, which may reflect the influence of different nonspecific neurotransmitters (*Nitz and McNaughton, 2004*; *Wilson and McNaughton, 1993*). Moreover, spike transmission changes across the familiar and novel environments were predicted by the rate changes of interneurons but not pyramidal cells. This suggests that postsynaptic effects such as differences in interneuron depolarization levels contributed to the changes in spike transmission between different environments. In addition to nonspecific neuromodulator transmitters, presynaptic place cells that were specifically active in the novel environment may have caused changes in the interneuron depolarization as well. Although novel environments may not entirely reorganize pyramidal connections, spatial learning is able to do so (*Dupret et al., 2013*). During the course of spatial learning, some interneurons increase their rates while others decrease, which are accompanied by changes in spike transmission probabilities of monosynaptic pairs. However, these spike transmission changes depended on both pyramidal and interneuron rate changes. One common aspect of the light and the spatial learning-mediated monosynaptic spike transmission changes is that, in both cases, it took place in a familiar environment in which some of the place cells remapped their place fields. In *Dupret et al., 2013* paradigm, some cells altered their place fields to represent the changed goal locations while in our paradigm, some of the place cells whose in-field firing was inhibited by the light remapped their place fields (*Schoenenberger et al., 2016*).

Can rules derived from in vitro observations, or those seen in vivo during anesthesia through afferent stimulations explain our findings during behavior? Our interneurons were recorded in the CA1 pyramidal layer where $Ca^{++}$ permeable AMPA receptors mediate the primary form of LTP and LTD. This form of plasticity requires the stimulation of their synaptic inputs and the hyperpolarization or a non-depolarized state of the interneuron (*Le Roux et al., 2013*; *Nissen et al., 2010*). In addition, metabotropic glutamate receptors can further regulate dendritic $Ca^{++}$ levels and the direction of synaptic plasticity (*Camiré and Topolnik, 2014*). In our case, the firing rate alteration of the interneurons was the strongest predictor of spike transmission change. The firing rate of the interneuron during FAML may reflect both the afferent excitation level of the cell as well as the light-induced inhibition; both of which could contribute to plastic changes. Excitability changes caused by the light stimulation are also expected to contribute to our observed results because both FAML-FAM1 and FAM2-FAM1 rate changes independently predicted FAM2-FAM1 spike transmission changes. It has been shown that Schaffer collateral stimulation enhances the excitability of CA1 PV cells following the stimulation, mediated via mGluR5 receptors (*Campanac et al., 2013*). In our dataset, the majority of interneurons showed a reduction in firing rate. Therefore, interneurons may be able to undergo both up- and downregulation of their excitation levels, which are not exclusively controlled by Schaffer collateral inputs.

We also saw that spike pairing of the pyramidal cells and interneuron in 50 ms time windows weakly influenced spike transmission, independent of the light-induced rate changes of these cells. Indeed, parvalbumin-positive CA1 interneurons exhibit NMDA-dependent associative plasticity as well (*Le Roux et al., 2013*) on their feedback connections from CA1 pyramidal cells. This may explain our spike pairing results. In vivo work suggested that theta-frequency afferent stimulation was optimal to induce LTP or LTD-like changes (*Lau et al., 2017*). Our light application occurred during theta oscillations, where such afferents would indeed provide theta-rhythmic stimulation of the CA1 interneurons. However, such a pairing relationship was observed only for a 50 ms time window. Moreover, it was no longer significant when pyramidal and interneuron rate changes were together taken into account. Because our optogenetic manipulation altered rates of individual cells without specifically influencing pyramidal-interneuron spike pairings, the combination of pyramidal and interneuron rates strongly predicted spike pairing probability. This can explain why spike pairing no longer predicted spike transmission changes when these rates were taken into account. After all, this result shows that rate-predicted spike pairing numbers are as good as the real ones to predict spike transmission changes. Nevertheless, we cannot exclude the possibility that the independent rate alterations of pyramidal cells and interneurons in FAML governed spike transmission probability changes, without spike pairing itself directly influencing it. Future work in which interneuron (or a certain genetic type) firing rate is selectively altered by optogenetics may provide further evidence for the independent contribution of postsynaptic interneuron depolarization in plasticity. Nevertheless, even in such manipulations, indirect alteration of pyramidal rates (e.g., because of disinhibition) is expected to occur.

Overall, our data indicate that during active behavior, changes in interneuron excitability that is coupled with spike pairing or altered presynaptic pyramidal spiking during theta epochs may trigger plasticity at the excitatory inputs to CA1 interneurons. Learning and the associated reorganization of the CA1 network may be a condition where such changes occur naturally.

## Materials and methods

**Key resources table**

| Reagent type (species) or resource | Designation | Source or reference | Identifiers | Additional information |
| --- | --- | --- | --- | --- |
| Strain, strain background (*Rattus norvegicus*) | *Long-Evans Rats* | Janvier, France | RRID:RGD-631593 | |
| Recombinant DNA reagent (Rattus norvegicus) | AAV2/1.CAMKII.ArchT.GFP.WPRE.SV40 | Penn Vector Core | RRID:Addgene: 26971-AAV1 | |

*Continued on next page*

*Continued*

| Reagent type (species) or resource | Designation | Source or reference | Identifiers | Additional information |
|---|---|---|---|---|
| Transfected construct | AAV2/1.CaMKIIα::eNpHR3.0-YFP | Penn Vector Core | RRID:Addgene: 99039-AAV1 | |
| Software, algorithm | Python | Python | RRID:SCR_008394 https://www.python.org | |
| Software, algorithm | LFP Online | GtiHub | https://github.com/igridchyn/lfp_online | |
| Other | 12 um tungsten wires | California Fine Wire | M294520 | |
| Other | Headstage amplifier | Axona, St. Albans, UK | http://www.axona.com | |

This study used previously published electrophysiological spike data (*Schoenenberger et al., 2016*). Accordingly, the experimental and spike clustering work has been described in this previous work in detail. Data from one additional rat recorded in the same paradigm and analyzed using the same methods was included in the data set.

## Surgery for virus injection and microdrive implantation

Four male adult rats (Long Evans, 300–500 g) were injected with a recombinant adeno-associated virus to express Halorhodopsin-YFP in the dorsal CA1 area (AAV2/1.CaMKIIα::eNpHR3.0-YFP *Zhang et al., 2007*, obtained from the Penn Vector Core facility, $1.6 \times 10^{13}$ genome copies/mL; Addgene 26971) and the remaining animal was injected with a recombinant adeno-associated virus to express Archaerhodopsin (ArchT) in the dorsal CA1 area (AAV2/1.CaMKII::ArchT.GFP.WPRE.SV40 (*Boyden et al., 2005*), obtained from the Penn Vector Core facility, $6.41*10^{12}$ genome copies per mL). The virus was injected at four sites into dorsal CA1 of the right hemisphere in four rats and bilaterally in one rat: site 1: −3.0 AP, ±2.2 L, 2.1 DV; site 2: −3.7 AP, ±2.9 L, 2.0 DV; site 3: −4.3 AP, ±3.5 L, 2.0 DV; site 4: −5.0 AP, ±4.2 L, 2.2 DV. 3.5 weeks after virus injection, animals were implanted with 15 (28 in one rat) independently movable wire-tetrodes under deep anesthesia using isoflurane (0.5–2%), oxygen (1–2 L/min) and an initial dose of buprenorphine (0.1 mg/kg). Tetrodes were attached to the 15-tetrode (24-tetrode and 4-octode in one rat) microdrive assemblies, enabling their independent movement. The tetrodes were constructed from four individual tungsten wires, 12 µm in diameter (H-Formvar insulation with Butyral bond coat, California Fine Wire, Grover Beach CA), twisted and then heated to bind them into a single bundle. The tips were then gold plated to reduce their impedance to 200–300 kΩ.

A 200 µm/0.48 NA optic fiber stub (Doric Lenses) located in the center of the tetrode array was used to apply laser light directly to the dorsal CA1 area. During surgery, a craniotomy was prepared above the dorsal hippocampus centered at AP = −4.0; ML = ± 3.0. Two stainless steel screws inserted through the skull above the cerebellum served as ground and reference electrodes, and six additional screws were used to permanently attach the microdrive assembly to the skull. Implantation was performed such as to position the tip of the optic fiber at a depth of 1.7 mm. The paraffin-wax coated electrodes and microdrives were then daubed with bone cement to encase the electrode-microdrive assembly and anchor it to the screws in the skull. Following a recovery period of 7 days, the tetrodes were lowered to their target locations over a further period of around 7 d. Tetrode locations were identified by electrophysiological markers such as theta band power, sharp wave polarity, and the presence of ripple oscillations, and by extrapolating the location of the electrodes by tracing the distances back along each electrode tract according to the daily advancement of the recorded electrodes. Implanted animals were housed individually in a separate room under a 12 hr light/12 hr dark cycle with ad libitum access to water, and they were maintained in a food-deprived state between 85–90% (plus an incremental 5 g per week) of their post-operative weight. Experiments were performed during the light phase. All rats used in this study were naïve and not used for additional procedures before surgery.

All procedures involving experimental animals were carried out in accordance with Austrian (Austrian Federal Law for experiments with live animals) animal law under a project license (BMBWF-66.018/0015-WF/V/3b/2014, BMBWF-66.018/0018-WF/V/3b/2019) approved by the Austrian Federal Science Ministry (BMWFW).

## Data acquisition

32-channel unity-gain preamplifier panels (Axona Ltd, St Albans, Hertfordshire, UK) were used to reduce cable movement artifacts. Wide-band (0.1/1 Hz – 5 kHz) recordings were taken, and the amplified local field potential and multiple-unit activity were continuously digitized at 24 kHz using a 128-channel data acquisition system (Axona Ltd, St Albans, Hertfordshire, UK). Two red LEDs mounted on the preamplifier headstage were used to track the location of the animal.

Green/yellow laser light for NpHR activation was provided by a 561 nm DPSS laser system equipped with an acousto-optic modulator (Omicron). The light was coupled into an optic fiber (four optic fibers in one rat) connected to a fiber-optic rotary joint (Doric lenses) from where a 200 µm/ 0.48 NA patch cord transmitted the light to the microdrive. Laser intensity was set to reach 25 mW total power at the tip of every implanted fiber stub. Data were recorded 6–7 weeks after AAV injection to ensure sufficient NpHR-YFP/ArchT GFP expression levels.

## Behavioral paradigms

Data was recorded while the animals explored different arenas or rested in a sleep box. The sleep box was small (20 cm × 27 cm) with 60 cm high walls and cushioned with a terry towel for the animal to sleep/rest comfortably. During training and electrode positioning, the animals were familiarized with a 120 cm circular environment with 20 cm high walls (minimum of 60 min of exposure per day for at least seven days) that served as the familiar arena in all experiments. Curtains were used to enclose this arena and provide a stable set of external cues. In all exploration sessions, small food pellets were dropped at random from an automated overhead system (2–3/min) to motivate the animals to explore the entire arena. For recordings in a novel environment, several other arenas with different sizes, shapes, and textures were used. In addition, curtains were opened to provide novel distal room cues.

Typical recording days consisted of 10 sessions: four exploration sessions flanked by five sleep sessions and a final test session where brief laser pulses were applied while the animal still rested in the sleep box. Typically, sleep and exploration sessions lasted 25 min, whereas the laser test session lasted 18 min. The animals first explored the familiar arena. After visiting a different novel arena, the familiar arena was explored again, but laser illumination was automatically triggered when the animal entered a specific part of the arena (light zone). Finally, the same arena was explored again. All exploration sessions were flanked by sleep. The light zone was defined by a center position and an angle between 120° and 180° such that it covered one-third to half of the arena. The initial angle defining the illumination zone was random and thus random also with respect to the hippocampal place fields. Every day, a novel illumination zone that had about 50% overlap with the previous day's zone was defined. During the course of the project and also within individual animals, the angle defining the size of the illumination zone was increased to include more place fields in the light zone. After completion of the experiments, the rats were deeply anesthetized and perfused through the heart first with PBS followed by a 4% buffered formalin phosphate solution for the histological verification of electrode tracks and optic fiber position. Furthermore, NpHR-YFP/ArchT-GFP expression in dorsal CA1 was verified in each animal by checking the fluorescence of the YFP/GFP tag.

## Spike sorting and unit classification

Unit isolation and clustering procedures have been described before (*Csicsvari et al., 1998*). Briefly, after resampling of the raw data to 20 kHz, action potentials were extracted from the digitally high-pass filtered (0.8–5 kHz) signal. The power computed in a sliding window (12.8 ms) and action potentials with a power of >5 SD from the baseline mean were selected. The spike features were then extracted using principal components analyses. The detected action potentials were then segregated into putative multiple single units using an automatic clustering software (*Harris et al., 2000*) (http://klustakwik.sourceforge.net/). Finally, the generated clusters were manually refined by a graphical cluster cutting program. Only units with clear refractory periods in their autocorrelation and well-defined cluster boundaries were used for further analysis. Periods of waking spatial exploration, immobility, and sleep were clustered together. The stability of the cells was verified by plotting spike features over time. In addition, an isolation distance (based on Mahalanobis distance, [*Harris et al., 2000*]) was calculated to ensure the spike clusters did not overlap during the recordings. CA1 pyramidal cells and interneurons were discriminated by their autocorrelations, firing rate,

and waveforms (*Csicsvari et al., 1999*; *Henze et al., 2000*). In total, we recorded 1842 pyramidal cells and 91 interneurons.

## Pyramidal cell-interneuron coupling

Isolation of monosynaptically-connected pyramidal cell-interneuron pairs was performed as described previously by identifying cross-correlograms between pyramidal cells and interneurons that exhibited a large, sharp peak in the 0.5–2.5 ms bins (after the discharge of the reference pyramidal cells) (*Csicsvari et al., 1998*). Because the number of action potentials used for the construction of these cross-correlograms varied from cell to cell, the histograms were normalized by dividing each bin by the number of reference pyramidal spike events (*Csicsvari et al., 1998*). The connection strength was thus accessed by measuring the spike transmission probability at the monosynaptic peak indicating the probability that the pyramidal cell would discharge its postsynaptic interneuron partner. However, the chance probability of the two cells firing together was subtracted in order to account for firing rate change-related fluctuations in the correlation strength. The chance firing probability was estimated by averaging the 10–50 ms bins on both sides of the histogram. The significance level for the monosynaptic peak was set at three standard deviations from the baseline ($p < 0.000001$) (*Abeles, 1982*; *Csicsvari et al., 1998*). In addition, to filter out sparse histograms, we only considered monosynaptic pairs in which either FAM1 or FAM2 contained at least 1000 spike coincidence counts with the −50 to 50 ms intervals, and the SD of the bin values were less than one-third of the mean bin value.

## Comparison of firing rate and firing field analysis

To compare firing rates between two sessions, we calculated the relative firing rate change by dividing the signed difference between the mean firing rates by the sum of the mean rates (i.e. $c = (r2-r1)/(r2+r1)$), where r1 and r2 denote the mean firing rates in the two sessions that are compared (*Leutgeb et al., 2004*; *Leutgeb et al., 2005*). This score is always between −1 and 1, and the extreme values −1 and 1 mean that a neuron is firing exclusively in one of the two sessions. A similar measure was used to measure the relative change of spike transmission and spike pairing events across different sessions.

## Statistical analyses

We used linear mixed models and ANOVA to determine the significance of variables in predicting spike transmission probabilities and their changes. We used a mixed model comparison to test whether predictions were independent of other variables and displayed corresponding partial correlations in the figures. We added the animal variable as a random effect to all linear mixed models to account for variability across animals and used the comparison of linear mixed model and linear model without an animal variable to test whether an animal variable contributed significantly to the prediction of spike probability and their changes. We used Holm-Bonferroni multiple testing correction to account for comparisons in multiple time windows in the analysis with prediction through pairing.

## Acknowledgements

We thank Michele Nardin and Federico Stella for comments on an earlier version of the manuscript. K Deisseroth for providing the pAAV-CaMKIIα::eNpHR3.0-YFP plasmid through Addgene. E Boyden for providing AAV2/1.CaMKII::ArchT.GFP.WPRE.SV40 plasmid through Penn Vector Core. This work was supported by the Austrian Science Fund (I02072 and I03713) and a Swiss National Science Foundation grant to PS. The authors declare no conflicts of interest.

## Additional information

### Funding

| Funder | Grant reference number | Author |
|---|---|---|
| Austrian Science Fund | I02072 | Jozsef Csicsvari |

| Swiss National Science Foundation | | Philipp Schoenenberger |
|---|---|---|
| Austrian Science Fund | I03713 | Jozsef Csicsvari |

The funders had no role in study design, data collection and interpretation, or the decision to submit the work for publication.

### Author contributions

Igor Gridchyn, Software, Formal analysis, Validation, Investigation, Writing - review and editing; Philipp Schoenenberger, Conceptualization, Software, Formal analysis, Funding acquisition, Investigation, Methodology, Project administration, Writing - review and editing; Joseph O'Neill, Conceptualization, Writing - review and editing; Jozsef Csicsvari, Conceptualization, Resources, Software, Formal analysis, Supervision, Funding acquisition, Methodology, Writing - original draft, Writing - review and editing

### Author ORCIDs

Igor Gridchyn https://orcid.org/0000-0002-1807-1929
Jozsef Csicsvari https://orcid.org/0000-0002-5193-4036

### Ethics

Animal experimentation: All procedures involving experimental animals were carried out in accordance with Austrian (Austrian federal Law for experiments with live animals) animal law under a project license (BMBWF-66.018/0015-WF/V/3b/2014, BMBWF-66.018/0018-WF/V/3b/2019) approved by the Austrian Federal Science Ministry (BMWFW).

### Decision letter and Author response

Decision letter https://doi.org/10.7554/eLife.61106.sa1
Author response https://doi.org/10.7554/eLife.61106.sa2

# Additional files

### Supplementary files

- Transparent reporting form

### Data availability

Original data and programs are available in the scientific repository of the Institute of Science and Technology Austria (https://doi.org/10.15479/AT:ISTA:8563).

The following dataset was generated:

| Author(s) | Year | Dataset title | Dataset URL | Database and Identifier |
|---|---|---|---|---|
| Csicsvari JL, Gridchyn I, Schönenberger P | 2020 | Optogenetic Alteration of Hippocampal Network Activity | https://doi.org/10.15479/AT:ISTA:8563 | IST Austria, 10.15479/AT:ISTA:8563 |

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
