## [Decision Letter]

**Acceptance summary:**

This study investigates into changes in pyramidal cell and interneuron spike transmission probability in pyramidal cell-interneuron monosynaptic pairs in hippocampal area CA1 following optogenetic inhibition and disinhibition of some of these cells in behaving rats exploring novel and familiar environments. Insights regarding how cell-cell correlations change with experience in freely behaving animals add to our understanding of how hippocampal circuits support learning and memory. The work also changes the current thinking about processes of plasticity in hippocampal networks and has relevance for interpretation of datasets in which optical inhibition is applied.

**Decision letter after peer review:**

Thank you for sending your article entitled "Optogenetic inhibition-mediated activity-dependent modification of CA1 pyramidal-interneuron connections during behavior" for peer review at *eLife*. Your article is being evaluated by three peer reviewers, and the evaluation is being overseen by a Reviewing Editor and Michael Frank as the Senior Editor.

The reviewers were all generally positive about the manuscript but also felt that it has a number of major issues in its current form. The lack of a YFP only control group was a major concern. Without an appropriate control group, the authors are unable to definitively state that their effects are due to optogenetic manipulations (e.g., such changes may just occur across the normal passage of time). However, the reviewers agreed in the consultation and discussion that they would not require the authors to include this optimal control group, assuming that the authors are able to demonstrate in some other way that the reported effects are actually due to light stimulation. One suggestion was offered by reviewer 2 (point 4). Another suggestion that came up in the Discussion was to compare the first and second halves of light-free sessions (i.e., FAM1 and FAM2). Another major concern was the lack of cell-type specificity of the CaMKII promoter (see reviewer 1's point 1 and reviewer 3's major point 2). Another essential revision is to demonstrate that the optogenetic manipulations did not cause behavioral changes that could potentially explain the differences in neuronal activity and co-activity patterns. Issues were also raised regarding statistical analyses (see reviewer 1's point 4 and other statistics-related points from reviewer 3). Lastly, the authors have previously shown the effects of these optogenetic manipulations on place cell responses. However, this was not entirely clear, and the authors should not assume that all readers have read their earlier paper (see points about place cells from reviewer 2 below). The authors should address the points raised by reviewer 2 by explicitly stating the effects that they have already shown and citing their earlier work. In addition, the authors should also discuss how the current plasticity results may be involved in the place cell changes reported in their earlier paper. The reviews are printed in their entirety below.

Reviewer #1:

The manuscript by Schoenenberger et al. all investigates into changes in pyramidal cell interneuron spike transmission probability in pyramidal cell-interneuron monosynaptic pairs in the rat hippocampal area CA1 following direct optogenetic inhibition and indirect disinhibition of some of these cells. The manuscript is a clever follow-up and re-analysis of the dataset collected for a paper that has already been published (Schoenenberger et al., 2016). The current analysis is focused at lasting changes of CA1 pyramidal cell-interneuron monosynaptic pairs following optogenetic inhibition. The authors report that these lasting changes are primarily predicted by changes in interneuron excitability during optogenetic modifications of firing rate and synchrony of hippocampal CA1 neurons.

This is an interesting study from one of the leading labs in the field and the reported changes in spike transmission probability between pyramidal cells and interneurons following optogenetic circuit manipulation are of potential importance for our understanding of activity-dependent reorganization of hippocampal circuits during learning.

There are some questions related to the analysis and to the interpretation of the data that should be addressed by the authors at this stage.

1) Similar to their previous paper (Schoenenberger et al., 2016), the authors' efforts here are somewhat also stymied by the apparent lack of cell-type specificity of the CaMKII viral promoter. This limitation should be explicitly stated and discussed in the current manuscript, especially since there are still conflicting results out there suggesting that these promoters could indeed be reliably used for cell-type specific targeting of principal cells, which is clearly not the case.

2) Relatedly, it is somewhat unfortunate that the optogenetic manipulation approach and the resulting dataset did not really allow the authors to directly test their main finding in the manuscript – that changes in interneuron excitably primarily dictates plasticity in their afferent inputs. This would have ideally required to selectively excite and/or inhibit interneurons in CA1. Of note, specific rAAV promoters actually do exist for selective manipulations of GABAergic interneurons (i.e., Dimidschstein et al., 2016).

3) The authors should analyze and report if the optogenetic manipulation has caused any acute (during FAML) or chronic (during FAM2) effects on the animals' behavior and to what extent these behavioral changes may contribute to the observed changes in pyramidal cell-interneuron spike transmission probability. On another note, the proper control for optogenetic experiments should have been the use of rAAV with a static fluorophore (i.e., YFP alone).

4) Statistical analysis should also be performed and reported on animal as a unit in order to better account for inter-animal variance.

Reviewer #2:

Schoenenberger et al. examine spike timing dynamics of pyramidal cell interneuron pairs in familiar and novel environments and following perturbation of pyramidal cell sub-populations in a familiar environment. The main findings address spike transmission probabilities for pyramidal cell interneuron pairs that are likely to have synaptic connections (based on short-latency cross-correlation peaks). Optogenetic inhibition of sub-populations of pyramidal neurons decreases activity in some neurons (as expected) and increases activity in others. Interneurons are largely inhibited in their activity during inhibition of pyramidal cells. Presumably, this is a result of reduced pyramidal cell excitation of feedback interneurons. Alterations in firing appear to produce longer-lasting changes in spike transmission probabilities. This suggests that spike timing relationships between pyramidal cells and interneurons are significant features of network dynamics that impact representation in hippocampus. In general, the findings are solidly backed up by the analytical approach and statistics and are clearly presented. There are some aspects of the data that seem absent from the report that might well make well place the results in a larger context and might make for greater impact.

1) The authors show that altered spike transmission probabilities in a novel environment are observed, but that these alterations do not impact dynamics in subsequent visits to the familiar environment. The authors do not reach far enough to attempt to explain this. Alterations induced by inhibition in the familiar environment do persist. These differences are interpreted as reflecting learning in the form of changes in pyramidal cell-interneuron synapses. This seems inconsistent with the lack of effect of novel environment dynamics. What might constitute an explanation for this? Perhaps there is some interaction with other inputs to interneurons that is critical? Furthermore, it would be of interest to determine whether it is the novelty of the environment that precludes persistence of spike transmission probabilities or simply the fact that the animal is in a different environment with a largely different set of pyramidal cell ensemble activity patterns.

2) The optogenetic inhibition was applied only in certain regions of the environment, yet the authors make no use of this design feature. Are the observed effects limited to pyramidal cell-interneuron pairs for which the pyramidal cell has a place field in the region of the environment where inhibition was applied? In general, a major limitation of the work is that it does not consider the effects of alterations in pyramidal cell-interneuron spike transmission on representation of place. Do the observed changes yield rate-remapping, global remapping, partial remapping, etc.? In the absence of such analyses, it is unclear whether one should consider the observed changes in interaction to be impactful on network function or not.

3) When considering the effects of co-activity, the authors should expand beyond the somewhat arbitrary time window of 20ms. It would be more informative to test a range of intervals and determine at what point paired spikes have no impact on subsequent transmission.

4) To place the results in context, the authors might include an analysis of odd versus even minutes of one or all of the inhibition-free sessions. This will provide somewhat of a baseline for spike transmission changes.

5) Do pyramidal cells released with increased firing in response to optogenetic inhibition of other pyramidal cells exhibit place-specific firing?

Reviewer #3:

In this manuscript, the authors intended to demonstrate that plastic changes occurred at the hippocampal pyramidal to interneuron connections in response to optogenetic manipulation during a spatial task in rats. To do so, the authors applied light stimulations as rats explored a familiar open arena (FAML session), after and before the rats explored the same arena without light in two sessions (FAM1, FAM2). The authors then compared spike transmission probability between identified putative pyramidal-interneuron pairs, as a measure of their connections, among these sessions. The authors made two key conclusions: (i) Optogenetic light stimulations led to lasting plasticity in spike transmission between connected pyramidal neurons and interneurons. (ii) The plastic changes were caused by firing rate changes in the postsynaptic interneurons. What and where synaptic plasticity occurs during behavior and how it leads to a particular learning behavior in vivo are important questions. However, identifying and manipulating synaptically connected neuron pairs is difficult in behaving animals. I applaud the authors' effort in this study and its outcome will be valuable to our understanding of learning and memory. However, as much as I like the study, I also have major concerns, which need to be resolved to make sure the key conclusions in the manuscript are valid.

1) Regarding the first conclusion, a major concern is that it is difficult to tell whether the observed changes had anything to do with the light stimulations. Judging from Figure 1C, D and Figure 2E, the changes in spike transmission and firing rate were small. One possibility is that these changes could arise just passively with time or other unrelated experience. What is lacking is a control experiment that includes the same recording procedure, but without the light stimulation or even better, with a control light stimulation session when neurons are designed not to respond (like a different color of light). I understand that this takes a lot of effort. However, at least in the existing data, the authors should analyze how spike transmissions within FAM1 or FAM2 fluctuate and how the fluctuation level was compared to the changes between FAM1 and FAM2.

2) I am confused by the authors' interpretation of the direct effect of light on firing rates of interneurons. The authors used halorhodopsin to inhibit neurons under the control of CaMKIIα. First, isn't it true that the promotor would restrict the halorhodopsin expression to pyramidal neurons, not much in interneurons? If this is not true, the authors need to provide references or histological evidence for this. Second, even if this is not true, how can the authors make sure the inhibition is caused directly by light, but not by the inhibition of pyramidal neurons? The evidence for a direct inhibition of interneurons (Figure 1B, for a light in sleep session) seems not strong, because the light was on for a long time. Third, some interneurons even increased their firing rates during FAML, which cannot be a direct effect of light. The picture of the light's direct effect is not clear in my mind, given a potential mixture of potential direct inhibition from light, if true, and the indirect inhibition from pyramidal neurons. The authors should clearly describe and discuss this issue, since this is important to their other key conclusion (see below).

3) The authors concluded that the plasticity in spike transmission was mainly caused by the light-induced changes in the firing rates of postsynaptic interneurons. First, how can the authors be sure that this is not the other way around? That is, plasticity could occur first, then led to reduced firing rates in interneurons? Second, the authors need to clearly state the direction of spike transmission plasticity caused by the interneuron rate change: lower/higher firing rates of interneuron lead to lower/higher spike transmissions. The authors touched the issue in the Discussion but seemed vague about this. One concern is that the enhanced firing rates in some interneurons were clearly not a direct effect of light stimulations, but they were important to the correlations the authors used to make the conclusion. I believe the authors should be straightforward about this and modify the conclusion accordingly.

4) One key result is Figure 3A. Here the FAML/FAM2 partial correlation was only reduced slightly from FAML alone, even though FAM2 alone was quite high. Can the authors verify and explain this?

5) The authors described the number of animals, the number of cells, and the number of pairs analyzed. It is unclear how many days were recorded, how many cells or pairs were obtained from each animal per day, whether at least the key results can be seen in multiple animals, and whether the same pairs were repeatedly used in the analysis.

6) What are the N's in Figure 2B-D, Figure 3A-B, Figure 4? What does each sample mean in these plots? If each sample was a day and there were 63 pairs in a number of days/animals, were there sufficient number of pairs for computing correlation or regression on a given day? What is the prediction in the labels? There is no description about this key analysis in the method.

7) What is the y-axis in Figure 1B?

[Editors' note: further revisions were suggested prior to acceptance, as described below.]

Thank you for submitting your article "Optogenetic inhibition-mediated activity-dependent modification of CA1 pyramidal-interneuron connections during behavior" for consideration by *eLife*. Your article has been reviewed by three peer reviewers, and the evaluation has been overseen by Laura Colgin as the Senior Editor and Reviewing Editor. The reviewers have opted to remain anonymous.

The reviewers have discussed the reviews with one another and the Reviewing Editor has drafted this decision to help you prepare a revised submission.

We would like to draw your attention to changes in our revision policy that we have made in response to COVID-19 (https://elifesciences.org/articles/57162). Specifically, when editors judge that a submitted work as a whole belongs in *eLife* but that some conclusions would benefit from additional new data or new experiments, as reviewers feel is the case with your paper, we are asking that the manuscript be revised to either limit claims to those supported by data in hand, or to explicitly state that the relevant conclusions would benefit from additional supporting data.

In the latter case, our expectation is that the authors will eventually carry out the additional experiments and report on how they affect the relevant conclusions either in a preprint on bioRxiv or medRxiv, or if appropriate, as a Research Advance in *eLife*, either of which would be linked to the original paper.

Summary:

This study investigates into changes in pyramidal cell interneuron spike transmission probability in pyramidal cell-interneuron monosynaptic pairs in hippocampal area CA1 following optogenetic inhibition and disinhibition of some of these cells in behaving rats exploring novel and familiar environments. Insights regarding how cell-cell correlations change with experience in freely behaving animals add to our understanding of how hippocampal circuits support learning and memory. The work also changes the current thinking about processes of plasticity in hippocampal networks and has relevance for interpretation of datasets in which optical inhibition is applied. However, the paper does include technical confounds, as noted below.

Figure 1C, D: the description of these two figure panels was not quantitative in both the main text and the figure legend. The authors stated that the plots "showed changes" and "altered the ability…", although the data points were highly correlated. It is better to quantify whether the changes were more than those in a control condition (like odd vs even minutes or two halves within FAM1). The authors' statements could be supported by the lower FAML-FAM1 correlations than those of the control condition. Also, it may be worth quantifying the overall changes (mean or median differences) in rate and transmission probability to understand whether a significant net effect of the stimulation occurred in this experiment.

Revisions expected in follow-up work:

1) Ideally, the proper controls for optogenetic experiments (i.e., YFP-only control) should have been included.

2) Follow-up experiments will include specific rAAV promoters for selective manipulations of GABAergic interneurons.

---

## [Author Response]

Reviewer #1:[…]There are some questions related to the analysis and to the interpretation of the data that should be addressed by the authors at this stage.1) Similar to their previous paper (Schoenenberger et al., 2016), the authors' efforts here are somewhat also stymied by the apparent lack of cell-type specificity of the CaMKII viral promoter. This limitation should be explicitly stated and discussed in the current manuscript, especially since there are still conflicting results out there suggesting that these promoters could indeed be reliably used for cell-type specific targeting of principal cells, which is clearly not the case.

Now we discuss the lack of cell specificity in the manuscript. Cell specificity of viruses using CaMKII promoter may depend on the region where the virus is expressed, and type of virus used. An earlier study has observed similar effects as well, e.g., Nathanson et al., 2009. In the Schoenenberger et al., 2016 paper we presented immunolabeling results showing that, in addition to pyramidal cells, our virus indeed expressed in both parvalbumin and somatostatin immunopositive cells and the light response delay of 1-2ms was similar for both the inhibited pyramidal cells and interneurons.

2) Relatedly, it is somewhat unfortunate that the optogenetic manipulation approach and the resulting dataset did not really allow the authors to directly test their main finding in the manuscript – that changes in interneuron excitably primarily dictates plasticity in their afferent inputs. This would have ideally required to selectively excite and/or inhibit interneurons in CA1. Of note, specific rAAV promoters actually do exist for selective manipulations of GABAergic interneurons (i.e., Dimidschstein et al., 2016).

Yes, unfortunately, in this manuscript, we were not able to selectively manipulate interneuron activity using new tools that have recently emerged. This finding was unexpected and selectively manipulating interneurons would require a collection of a dataset in size and effort similar to the one we use in this study. Note also that even in cases in which interneuron firing is specifically altered using optogenetics, such manipulations will indirectly influence the firing of pyramidal cells. For example, suppressing interneuron activity will lead to the disinhibition of pyramidal cells. We see such disinhibitory effect even when we selectively inhibit a smaller subset of CCK interneurons in transgenic mice.

3) The authors should analyze and report if the optogenetic manipulation has caused any acute (during FAML) or chronic (during FAM2) effects on the animals' behavior and to what extent these behavioral changes may contribute to the observed changes in pyramidal cell-interneuron spike transmission probability. On another note, the proper control for optogenetic experiments should have been the use of rAAV with a static fluorophore (i.e., YFP alone).

In the Schoenenberger et al., 2016 paper, we compared the speed across different sessions of the animal and possible occupancy differences in the light zone and outside. Now, we performed the same analysis in our extended data as well (Figure 1—figure supplement 1). No significant differences were seen either in occupancy or speed, inside and outside the light zone across FAM1, FAML and FAM2.

In relation to the control experiments, as suggested by the Editor’s letter, we assessed the within-session variability of spike transmission probabilities in FAM1 and showed that these are smaller than those seen across FAM1-FAM2 (Figure 2B).

4) Statistical analysis should also be performed and reported on animal as a unit in order to better account for inter-animal variance.

As suggested, in all analyses, we incorporated animal as a random effect in our linear mixed models and ANOVA and show that this did not influence our results.

Reviewer #2:[…]1) The authors show that altered spike transmission probabilities in a novel environment are observed, but that these alterations do not impact dynamics in subsequent visits to the familiar environment. The authors do not reach far enough to attempt to explain this. Alterations induced by inhibition in the familiar environment do persist. These differences are interpreted as reflecting learning in the form of changes in pyramidal cell-interneuron synapses. This seems inconsistent with the lack of effect of novel environment dynamics. What might constitute an explanation for this? Perhaps there is some interaction with other inputs to interneurons that is critical? Furthermore, it would be of interest to determine whether it is the novelty of the environment that precludes persistence of spike transmission probabilities or simply the fact that the animal is in a different environment with a largely different set of pyramidal cell ensemble activity patterns.

In the revision, we discuss further the possible mechanism behind the altered spike transmission in the novel environment. We show in Figure 4A that spike transmission changes from familiar to novel environments were predicted by the change of the interneuron rates but not by the pyramidal rates. This suggests a postsynaptic effect, which, at least in part, may be related to the change of the depolarization/excitability of interneurons. We agree with the reviewer that other inputs to the interneurons may be a cause for the spike transmission changes from the familiar to the novel environment. Such inputs may include other place cell or non-specific neuromodulation, e.g., acetylcholine levels are higher in a novel environment. Unfortunately, we do not have data in which two different familiar environments and a novel environment are all simultaneously recorded, which would be required if pyramidal-interneuron spike transmissions reorganize across familiar environments.

2) The optogenetic inhibition was applied only in certain regions of the environment, yet the authors make no use of this design feature. Are the observed effects limited to pyramidal cell-interneuron pairs for which the pyramidal cell has a place field in the region of the environment where inhibition was applied? In general, a major limitation of the work is that it does not consider the effects of alterations in pyramidal cell-interneuron spike transmission on representation of place. Do the observed changes yield rate-remapping, global remapping, partial remapping, etc.? In the absence of such analyses, it is unclear whether one should consider the observed changes in interaction to be impactful on network function or not.

In the previous Schoenenberger et al., 2016 paper, we showed that a fraction of place cells that were inhibited by the light remapped their place fields as a result of light-mediated inhibition. In a new analysis, we attempted to relate place field remapping to the changes of monosynaptic connections, but we did not see a relationship.

We report this in the revision by saying that “In addition to the changing the firing rate, light application can cause remapping in a subpopulation of cells (Schoenenberger et al., 2016). However, a change in spike transmission in FAM2 did not predict the degree of pyramidal place field remapping (P=0.1612, F-test)”.

We may not have been able to see such a relationship because only a subpopulation of light-inhibited cells exhibited place field remapping.

3) When considering the effects of co-activity, the authors should expand beyond the somewhat arbitrary time window of 20ms. It would be more informative to test a range of intervals and determine at what point paired spikes have no impact on subsequent transmission.

We checked the effect in additional time windows of 10ms, 20ms, 50ms and 100ms. When we included animal and used different intervals (leading to multiple comparisons) as additional factors, only the 50ms time window was significant. Moreover, spike paring no longer predicted the spike transmission changes when pyramidal and interneuron rate changes together were accounted for. This is possibly due to the fact that spike pairing alone during FAML is very strongly (R^2^=0.849) predicted by the combined rate changes (Figure 6B). We comment on these results in the Discussion now:

“However, such a pairing relationship was observed only for 50 ms time window. […] Nevertheless, we cannot exclude the possibility that the independent rate alterations of pyramidal cells and interneurons in FAML governed spike transmission probability changes, without spike pairing itself directly influencing it.”

4) To place the results in context, the authors might include an analysis of odd versus even minutes of one or all of the inhibition-free sessions. This will provide somewhat of a baseline for spike transmission changes.

We thank the reviewer for this suggestion, and we perform such an analysis as well to show stability. Indeed, in the difference in the spike transmission measured in alternating time windows within FAM1 was significantly less than across FAM1-FAM2 (Figure 2B).

5) Do pyramidal cells released with increased firing in response to optogenetic inhibition of other pyramidal cells exhibit place-specific firing?

Yes, in the Schoenenberger et al., 2016 paper, we showed that both disinhibited and inhibited pyramidal cells exhibited place-related firing. Interestingly, disinhibition did not alter the place fields, whereas inhibition triggered place field remapping.

Reviewer #3:[…]1) Regarding the first conclusion, a major concern is that it is difficult to tell whether the observed changes had anything to do with the light stimulations. Judging from Figure 1C, D and Figure 2E, the changes in spike transmission and firing rate were small. One possibility is that these changes could arise just passively with time or other unrelated experience. What is lacking is a control experiment that includes the same recording procedure, but without the light stimulation or even better, with a control light stimulation session when neurons are designed not to respond (like a different color of light). I understand that this takes a lot of effort. However, at least in the existing data, the authors should analyze how spike transmissions within FAM1 or FAM2 fluctuate and how the fluctuation level was compared to the changes between FAM1 and FAM2.

In the revision, as suggested by reviewer 2 as well, we compare spike transmissions within FAM1 using alternating time windows and compare it to the changes occurring across FAM1-FAM2 sessions. The within-session variability of spike transmission was significantly less than those across the FAM1-FAM2 sessions (Figure 2B). Note, however, that some additional findings of the manuscript also argue against the possibility that our effects are simply due to random changes that may occur over time passed. First, we showed that changes in the light session but not those in the novel session predict the changes across FAM1-to-FAM2 sessions. Second, we show that other factors such interneuron rate changes will also predict spike transmission changes. We do not see how random fluctuations of spike transmission could still lead to predictions that involve factors of the light session but not the other “control” intervening session of the novel environment.

2) I am confused by the authors' interpretation of the direct effect of light on firing rates of interneurons. The authors used halorhodopsin to inhibit neurons under the control of CaMKIIα. First, isn't it true that the promotor would restrict the halorhodopsin expression to pyramidal neurons, not much in interneurons? If this is not true, the authors need to provide references or histological evidence for this. Second, even if this is not true, how can the authors make sure the inhibition is caused directly by light, but not by the inhibition of pyramidal neurons? The evidence for a direct inhibition of interneurons (Figure 1B, for a light in sleep session) seems not strong, because the light was on for a long time. Third, some interneurons even increased their firing rates during FAML, which cannot be a direct effect of light. The picture of the light's direct effect is not clear in my mind, given a potential mixture of potential direct inhibition from light, if true, and the indirect inhibition from pyramidal neurons. The authors should clearly describe and discuss this issue, since this is important to their other key conclusion (see below).

We further discussed these issues in the revised manuscript. In the Schoenenberger et al., 2016 paper, we quantified these effects. We showed that both the light-inhibited pyramidal cells and interneurons suppressed their firing within a short 1-2ms time delay relative to the light onset. The light responses were tested using 500ms light pulses in the end of the recordings while the animal was rested. We also performed immunolabeling in the Schoenenberger et al., 2016 study and showed that both parvalbumin and somatostatin immunopositive cells expressed halorhodopsin. Note also that an earlier paper reported a similar effect Nathanson et al., 2009. Of course, we cannot exclude that this virus is more specific in other brain regions or other virus serotypes or constructs may be more specific.

3) The authors concluded that the plasticity in spike transmission was mainly caused by the light-induced changes in the firing rates of postsynaptic interneurons. First, how can the authors be sure that this is not the other way around? That is, plasticity could occur first, then led to reduced firing rates in interneurons? Second, the authors need to clearly state the direction of spike transmission plasticity caused by the interneuron rate change: lower/higher firing rates of interneuron lead to lower/higher spike transmissions. The authors touched the issue in the Discussion but seemed vague about this. One concern is that the enhanced firing rates in some interneurons were clearly not a direct effect of light stimulations, but they were important to the correlations the authors used to make the conclusion. I believe the authors should be straightforward about this and modify the conclusion accordingly.

Indeed, Figure 3C shows that the interneuron rate increase in FAML is associated with a stronger spike transmission, whereas in cases of reduced spike transmission, FAM2 interneuron rate is weaker. As asked, in the revision, we spelled out this relationship in the Discussion. We agree that the rate increase of some interneurons may be caused by strengthened pyramidal connections triggered by complex network effects mediated by light-induced inhibition of a subgroup of pyramidal cells and interneurons. We discussed this scenario in revision, as suggested by the reviewer.

4) One key result is Figure 3A. Here the FAML/FAM2 partial correlation was only reduced slightly from FAML alone, even though FAM2 alone was quite high. Can the authors verify and explain this?

This finding showed that, although FAM2-FAM1 interneuron rate changes predicted the corresponding spike transmission changes, similar rate changes between FAML-FAM1 have further predictive value. The relationship between FAM2-FAM1 rate and spike transmission changes may solely suggest that excitability changes of the interneuron led to the spike transmission changes. However, with the partial correlation (and the associated ANOVA model comparisons), we were able to show that rate (i.e., excitability) alterations in the light session were able to independently influence the FAM1-FAM2 spike transmission changes. That is that excitability/depolarization alterations in FAML that were no longer present in FAM2 still influenced FAM1-FAM2 spike transmission changes. We explain this better in the revision.

5) The authors described the number of animals, the number of cells, and the number of pairs analyzed. It is unclear how many days were recorded, how many cells or pairs were obtained from each animal per day, whether at least the key results can be seen in multiple animals, and whether the same pairs were repeatedly used in the analysis.

In the original work in four animals, we recorded from n=31, 23, 5, 4 detected cell pairs. In the revision, we added one additional animal with n=16 cell pairs to ensure that all the results could be replicated independently of the critical contribution of a single animal. In all analyses, we included animal as a random effect and showed that this did not influence our results. In three animals we recorded in two recording days, while four recording days were used in the remaining two animals. The electrodes were moved between recording days to ensure that a different set of cells are recorded across days. Therefore, we think that only very few cell pairs may have recorded across the two recordings days. Yet, even if we did record from the same cell pairs across days, the light zone location was changed daily; therefore, the cells experienced an entirely different network activity background in our manipulations with different potential outcomes.

6) What are the N's in Figure 2B-D, Figure 3A-B, Figure 4? What does each sample mean in these plots? If each sample was a day and there were 63 pairs in a number of days/animals, were there sufficient number of pairs for computing correlation or regression on a given day? What is the prediction in the labels? There is no description about this key analysis in the method.

In each plot, each dot represents the value related to a single cell pair so n=78 (extended dataset) always. With a few sessions, we were able to record >10 pairs yet the yield of detecting these monosynaptic pairs was relatively low. Prediction refers to the linear regression and the correlation coefficient (r). We added correlation to the legend. We thank the reviewer for pointing out the confusion. However, we kept prediction on the axes because prediction is easier to understand (according to our opinion) than using “correlation with” on these labels.

7) What is the y-axis in Figure 1B?

It shows the probability of a spike occurring within the 20ms time bins. Now we also explain it in the legend.

[Editors' note: further revisions were suggested prior to acceptance, as described below.]

Revisions for this paper:Figure 1C, D: the description of these two figure panels was not quantitative in both the main text and the figure legend. The authors stated that the plots "showed changes" and "altered the ability…", although the data points were highly correlated. It is better to quantify whether the changes were more than those in a control condition (like odd vs even minutes or two halves within FAM1). The authors' statements could be supported by the lower FAML-FAM1 correlations than those of the control condition. Also, it may be worth quantifying the overall changes (mean or median differences) in rate and transmission probability to understand whether a significant net effect of the stimulation occurred in this experiment.

We performed the requested quantification and indeed the within session (FAM1) correlations we higher that across session (FAM1-FAML) ones, both for firing rate and spike transmission. Median rates were not significantly different however because cells either reduced or increased their rate during the light application. But there was a significant reduction in the spike transmission probabilities from FAM1 to FAML.

Revisions expected in follow-up work:1) Ideally, the proper controls for optogenetic experiments (i.e., YFP-only control) should have been included.

We acknowledge this in the beginning of the Discussion saying that “Or study did not use control animals in which only YFP was expressed. Therefore, we cannot exclude the possibility that optogenetic channel expression, or, perhaps, light application enhanced the plasticity on pyramidal-interneuron synapses. Yet, we observed similar activity-dependent changes during spatial learning before (Dupret et al., 2013). So, it is likely that the optogenetic, light-mediated rate alteration was a primary driver of the activity-dependent, lasting connection-weight changes. “

2) Follow-up experiments will include specific rAAV promoters for selective manipulations of GABAergic interneurons.

We state these follow-up experiments in the Discussion but we also point out that the disinhibition of pyramidal cells might hinder these experiments: “Future work in which interneuron (or a certain genetic type) firing rate is selectively altered by optogenetics may provide further evidence for the independent contribution of postsynaptic interneuron depolarization in plasticity. Nevertheless, even in such manipulations, indirect alteration of pyramidal rates (e.g., because of disinhibition) is expected to occur.”